# HSVA: Hierarchical Semantic-Visual Adaptation for Zero-Shot Learning

**Shiming Chen[†], Guo-Sen Xie[‡], Yang Liu[§], Qinmu Peng[†], Baigui Sun[§],**
**Hao Li[§], Xinge You[†*], Ling Shao[‡♯]**
[†]Huazhong University of Science and Technology (HUST), China
[§]Alibaba Group, Hangzhou, China
[‡]Mohamed bin Zayed University of AI (MBZUAI), UAE
[♯]Inception Institute of Artificial Intelligence (IIAI), UAE
{shimingchen,youxg}@hust.edu.cn    ling.shao@ieee.org

## Abstract

Zero-shot learning (ZSL) tackles the unseen class recognition problem, transferring semantic knowledge from seen classes to unseen ones. Typically, to guarantee desirable knowledge transfer, a common (latent) space is adopted for associating the visual and semantic domains in ZSL. However, existing common space learning methods align the semantic and visual domains by merely mitigating distribution disagreement through one-step adaptation. This strategy is usually ineffective due to the heterogeneous nature of the feature representations in the two domains, which intrinsically contain both distribution and structure variations. To address this and advance ZSL, we propose a novel hierarchical semantic-visual adaptation (HSVA) framework. Specifically, HSVA aligns the semantic and visual domains by adopting a hierarchical two-step adaptation, i.e., structure adaptation and distribution adaptation. In the structure adaptation step, we take two task-specific encoders to encode the source data (visual domain) and the target data (semantic domain) into a structure-aligned common space. To this end, a supervised adversarial discrepancy (SAD) learning is proposed to adversarially minimize the discrepancy between the predictions of two task-specific classifiers, thus making the visual and semantic feature manifolds more closely aligned. In the distribution adaptation step, we directly minimize the Wasserstein distance between the latent multivariate Gaussian distributions to align the visual and semantic distributions using a common encoder. Finally, the structure and distribution adaptation are derived in a unified framework under two partially-aligned variational autoencoders. Extensive experiments on four benchmark datasets demonstrate that HSVA achieves superior performance on both conventional and generalized ZSL. The code is available at https://github.com/shiming-chen/HSVA .

## 1 Introduction

Over the past decade, significant progress has been made in zero-shot learning (ZSL), which aims to recognize new classes during learning by exploiting the intrinsic semantic relatedness between seen and unseen categories [1, 2, 3]. Inspired by the way humans learn unknown concepts, side-information (e.g., attributes [4], word vectors [5], and sentences [6]) related to seen/unseen classes is employed to guarantee knowledge transfer between the seen/unseen data. Common space learning is one typical method for representing the relationship between the visual and semantic domains for ZSL, which is key to dealing with the knowledge transfer [7]. However, existing common space

---

[*]Corresponding author

35th Conference on Neural Information Processing Systems (NeurIPS 2021), virtual.

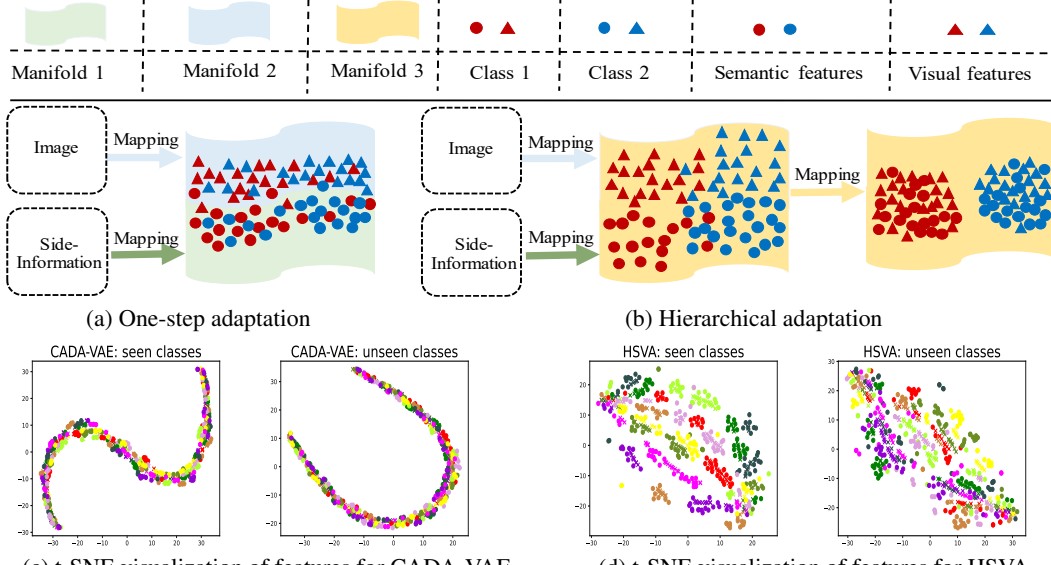

(a) One-step adaptation       (b) Hierarchical adaptation

(c) t-SNE visualization of features for CADA-VAE       (d) t-SNE visualization of features for HSVA

Figure 1: Illustration of HSVA. Common space learning methods learn domain-aligned feature representations for semantic and visual domains in latent embedding space with semantic-visual adaptation. However, the heterogeneous feature representations of the semantic and visual domains vary in both distribution and structure. (a) One-step adaptation focuses on distribution alignment between visual and semantic domains, neglecting structure variation. This causes the semantic and visual distributions to be located in different manifolds, resulting in the misclassification of some samples. (b) Hierarchical adaptation, in contrast, can learn an intrinsic and discriminative common space for semantic and visual feature representations by adopting sequential structure adaptation and distribution adaptation. We provide t-SNE visualizations [14] of features learned by (c) CADA-VAE [11] and (d) our HSVA on 10 classes from AWA2 (the "o" and "x" indicate visual and semantic features, respectively, and different colors denote different classes). Best viewed in color.

learning methods align the distributions of the semantic and visual domains with one-step adaptation [8, 9, 7, 10, 11, 12, 13], while neglecting the fact that the heterogeneous feature representations of distinct semantic and visual domains have both distribution and structure variations. Thus, their performance is limited. In this work, we propose a novel common space learning formulation that leverages a hierarchical semantic-visual adaptation to learn an intrinsic common space for a better alignment of the two heterogeneous representation domains.

The heterogeneous feature representations of the semantic and visual domains are distinct [15, 16]. Although one-step adaptation learns distribution-aligned feature representations for the semantic and visual domains using two different mapping models (e.g., encoders) and distribution alignment constraints (e.g., maximum mean discrepancy (MMD)), the two distributions are located in different manifolds since the one-step adaptation does not consider the manifold structure relationship between visual and semantic representations, as shown in Figure 1 (a) and (c). If we take the Euclidean distance or manifold distance [17, 18] to measure the relationship between different classes, the classifier inevitably misclassifies some samples, thus leading to inferior ZSL performance. As such, mapping the semantic and visual domains into an intrinsic and desirable common space for conducting an effective ZSL is highly necessary.

To learn structure- and distribution-aligned feature representations for distinct visual and semantic domains, we propose a *hierarchical semantic-visual adaptation* (HSVA) framework to learn an intrinsic common space that contains the essential multi-modal information associated with unseen classes (Figure 1 (b) and (d)). HSVA aligns the semantic and visual domains with structure adaptation (SA) and distribution adaptation (DA) in a unified framework, under two partially-aligned variational autoencoders. In the SA, we use two task-specific encoders to encode source data (visual domain) and target data (semantic domain), respectively, into a structure-aligned space, which is learned through supervised adversarial discrepancy (SAD) learning to minimize the discrepancy between

the predictions of the two task-specific classifiers. This encourages the visual and semantic feature manifolds to be closer to each other, thus aligning the manifold structure variation. In the DA, we map the structure-aligned features into a distribution-aligned common space using a common encoder, which is optimized by minimizing the Wasserstein distance between the latent multivariate Gaussian distributions of the visual and semantic domains. The common encoder preserves the structure-aligned representations in the DA. Thus, HSVA can learn a structure and distribution-aligned common space for visual and semantic feature representations, which is better than the distribution-aligned feature representations learned by existing common space learning methods [8, 9, 7, 10, 11, 19]. To the best of our knowledge, this is the first work that leverages hierarchical semantic-visual adaptation to address heterogeneous feature alignment in ZSL. Following [11], we conduct extensive experiments on four challenging benchmark datasets, i.e., CUB, SUN, AWA1, and AWA2, under both conventional and generalized ZSL settings. The proposed HSVA achieves consistent improvement over the existing common space learning methods on all datasets. We show qualitatively that the structural variation is important for the interaction between visual and semantic domains. Moreover, we show significantly better common space for visual and semantic feature representations than the existing common space learning methods, e.g., CADA-VAE [11].

## 2   Related Work

**Zero-Shot Learning.**    In practice, the relationship between visual and semantic domains is determined by learning an embedding space where semantic vectors and visual features interact [20, 21, 22, 23, 24, 25, 26, 19]. There are three mapping methods for learning such an embedding space, including direct mapping, model parameter transfer, and common space learning [7]. Direct mapping learns a mapping function from visual features to semantic representations [4, 20, 27]. However, its generalization is limited by the high intra-class variability of the visual domain. Meanwhile, the lower-dimensional semantic space shrinks the variance of the projected data points, resulting in the hubness problem [28]. Model parameter transfer, in contrast, takes place in the visual space, where the model parameters for unseen classes are usually obtained [29, 30]. In essence, generative ZSL follows this methodology, simultaneously learning semantic→visual mapping and data augmentation [22, 31, 32, 22, 33]. However, since the inter-class relationships among unseen classes are not taken into account, model parameter transfer is often limited [7]. Finally, common space learning learns a common representation space into which both visual features and semantic representations are projected for effective knowledge transfer [34, 9, 7, 10, 11]. By calibrating the common space using bidirectionally aligned knowledge of the visual and semantic representations, it can simultaneously avoid the issues of direct mapping and model parameter transfer. As such, common space learning is a promising methodology for ZSL. However, existing common space learning methods align the semantic and visual distributions with one-step adaptation, which neglects the distinct heterogeneous feature representations of the semantic and visual domains that are characterized by structure and distribution variations. This causes the distribution-aligned common space to be located in a different manifold, and samples to thus inevitably misclassified. In contrast, we propose an effective framework with both structure adaptation and distribution adaptation to address these challenges.

**Domain Adaptation.**    Domain adaptation aims to learn domain-invariant representations by minimizing the discrepancy between the distributions of source and target via distribution alignment, domain adversarial learning, or task-specific methods. It can be divided into homogeneous and heterogeneous domain adaptation according to the characteristics of the source and target data. In the homogeneous domain adaptation setting, the feature spaces of the source and target domains are identical, with the same dimensions. Thus, one-step adaptation can be employed [35, 36, 37]. In contrast, in the heterogeneous setting, the feature spaces of the source and target domains are distant, and their dimensions also typically differ. Since there is little overlap between the two domains, one-step domain adaptation is not effective in the heterogeneous domain adaptation [38, 39]. Thus, multi-step (or transitive) domain adaptation methods are used for heterogeneous domain adaptation. However, these methods cannot be directly used in ZSL. Since ZSL has four domains (i.e., seen, unseen, semantic, and visual), knowledge cannot be transferred between two domains only, i.e., the distributions of the source and target domains cannot be aligned. We argue that the structure variation should also be taken into account when aligning the visual and semantic domains in ZSL.

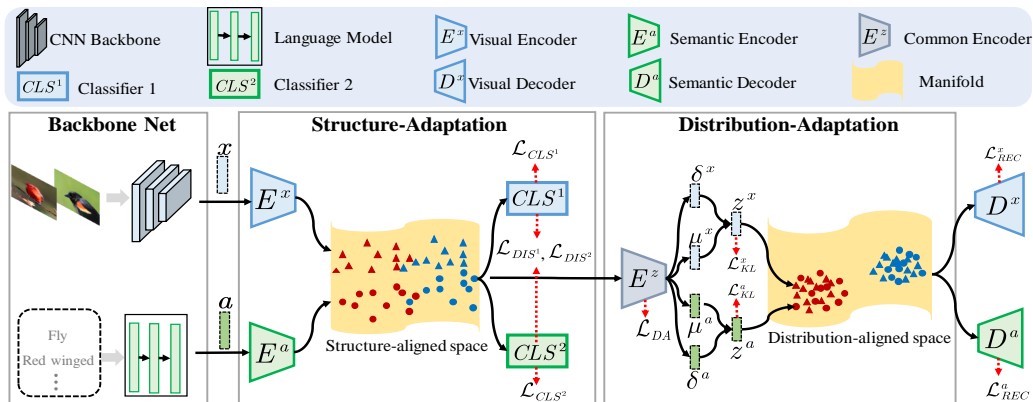

Figure 2: The framework of the proposed hierarchical semantic-visual adaptation (HSVA). HSVA consists of two partially-aligned variational autoencoders, which simultaneously perform structure adaptation and distribution adaptation with various constraints. Our two-step adaptation formulation helps HSVA learn structure and distribution-aligned feature representations for visual and semantic in an unified framework, which avoids the adverse effects of heterogeneous feature representations between distinct visual and semantic domains.

## 3 Hierarchical Semantic-Visual Adaptation

The task of ZSL is formulated as follows. Let $S = \{(x^s, y^s, a^s) \mid x^s \in X, y^s \in Y^s, a^s \in \mathcal{A}\}$ be a training set consisting of image features $x$ extracted by a convolutional neural network (CNN) Backbone (e.g., ResNet-101), where seen class embeddings $a^s$ are learned by a language model and seen class labels $y^s$. In addition, an auxiliary training set $U = \{(y^u, a^u) \mid y^u \in Y^u\}$ from unseen classes is used, where $y^u$ denotes unseen classes from a set $Y^u$, which is disjoint from $Y^S$. $x^u \in X$ are the unseen visual features in the test set. Note that $x = x^s \cup x^u$ and $a = a^s \cup a^u$. The goal of conventional ZSL (CZSL) is to learn a classifier for unseen visual features, i.e., $f_{CZSL} : X \rightarrow Y^U$, while for generalized ZSL (GZSL) the goal is to learn a classifier for seen and unseen visual features, i.e., $f_{GZSL} : X \rightarrow Y^U \cup Y^S$.

In the following, we introduce our HSVA, which learns an intrinsic common space for semantic and visual feature representations to conduct ZSL. As shown in Figure 2, our framework consists of two partially-aligned autoencoders, which conduct structure adaptation (SA) and distribution adaptation (DA). In the end of this section, we demonstrate how we perform zero-shot learning using the feature representations extracted from the intrinsic common manifold space.

### 3.1 Partially-Aligned Variational Autoencoders

Our HSVA model consists of two partially-aligned variational autoencoders. The first includes two feature encoders (visual encoder $E^x$ and common encoder $E^z$) and one decoder (visual decoder $D^x$). $E^x$ and $E^z$ are used to learn structure and distribution-aligned visual embeddings, respectively, which share two common spaces (i.e., structure and distribution-aligned spaces) with the semantic embeddings. $D^x$ aims to decode the distribution-aligned embeddings from the visual and semantic features to be similar to the visual features. Analogously, the second variational autoencoder has a similar structure to the first (i.e., it includes a semantic encoder $E^a$, the $E^z$, and a semantic decoder $D^a$), but it is employed to learn semantic embeddings. The partially-aligned variational autoencoders are first optimized by two variational autoencoders (VAE) losses:

$$\mathcal{L}_{VAE}^x(E^x, E^z) = \mathbb{E}[\log D^x(z^{x^s})] - \gamma \text{KL}(E^z(E^x(x^s))\|p(z^{x^s}|x^s)), \tag{1}$$

$$\mathcal{L}_{VAE}^a(E^a, E^z) = \mathbb{E}[\log D^a(z^{a^s})] - \gamma \text{KL}(E^z(E^a(a^s))\|p(z^{a^s}|a^s)), \tag{2}$$

where $z^{x^s}$ and $z^{a^s}$ are the visual and semantic embeddings in the distribution-aligned space, $\gamma$ is the weight of the KL-divergence, and $p(z^{a^s}|a^s)$ and $p(z^{x^s}|x^s)$ are the prior distributions assumed to be $\mathcal{N}(0, 1)$. To make the visual and semantic embeddings in the structure- and distribution-aligned spaces more consistent, we then use a cross reconstruction loss to constrain the two partially-aligned

variational autoencoders:

$$\mathcal{L}_{REC}(E^x, E^a, E^z, D^x, D^a) = \mathcal{L}^x_{REC}(E^x, E^z, D^x) + \mathcal{L}^a_{REC}(E^a, E^z, D^a) \quad (3)$$

where

$$\mathcal{L}^x_{REC}(E^x, E^z, D^x) = \left\| x^s - D^x(z^{a^s}) \right\|_1, \quad (4)$$

$$\mathcal{L}^a_{REC}(E^a, E^z, D^a) = \left\| a^s - D^a(z^{x^s}) \right\|_1. \quad (5)$$

Finally, the two partially-aligned variational autoencoders conduct structure and distribution adaptation for the visual and semantic features, which will be introduced in the following.

## 3.2 Structure Adaptation

Structure adaptation (SA) is used to guide the encoder $E^x$ and $E^a$ to learn a structure-aligned space for the visual and semantic features. Previous methods [7, 11] directly align the semantic and visual features with distribution distance constraints (e.g., MMD), which neglects the structure variation between the heterogeneous feature representations of the two domains. Furthermore, these methods fail to learn discriminative features as they do not take category relationships into account, as shown in Figure 1 (a). Motivated by task-specific unsupervised domain adaptation [36, 37], we propose a supervised adversarial discrepancy (SAD) learning to minimize the discrepancy between the outputs of two task-specific classifiers ($CLS^1$, $CLS^2$). This enables us to learn a class decision boundary and embed semantic and visual features into a structure-aligned common space for SA. Thus, SA can learn discriminative structure-aligned representations for semantic and visual features. Different from [36, 37], which only detect the target samples (e.g., semantic embeddings $E^a(a)$) to learn a single encoder (generator), SAD simultaneously detects the source samples (e.g., visual embeddings $E^x(x)$) and target samples (e.g., semantic embeddings $E^a(a)$) to learn a good structure-aligned common space with two task-specific encoders. Specifically, SAD includes three steps for optimization: 1) semantic and visual embedding classification for $E^x$, $E^a$, $CLS^1$ and $CLS^2$; 2) discrepancy maximization for $CLS^1$ and $CLS^2$; and 3) discrepancy minimization for $E^x$ and $E^a$.

**Semantic and Visual Embedding Classification.** We train $E^x$, $E^a$, $CLS^1$ and $CLS^2$ by minimizing the softmax cross-entropy to collectly classify the structure-aligned semantic and visual embeddings. This is crucial for enabling classifiers and encoders ($E^x$, $E^a$) to obtain task-specific discriminative features. The visual and semantic classification loss is formulated as:

$$\mathcal{L}_{CLS}(E^x, E^a, CLS^1, CLS^2) = \mathcal{L}_{CLS^1}(E^x(x^s), y^s) + \mathcal{L}_{CLS^2}(E^x(x^s), y^s) \\ + \mathcal{L}_{CLS^1}(E^a(a^s), y^s) + \mathcal{L}_{CLS^2}(E^a(a^s), y^s), \quad (6)$$

where

$$\mathcal{L}_{CLS^1}(\mathbf{x}, \mathbf{y}) = \mathcal{L}_{CLS^2}(\mathbf{x}, \mathbf{y}) = -\mathbb{E} \sum_{k=1}^{K} \mathbb{I}_{[k=\mathbf{y}]} \log p\left(y \mid \mathbf{x}\right). \quad (7)$$

Here K is the number of seen classes, $\mathbb{I}_{[k=\mathbf{y}]}$ is an indicator function (i.e., it is one when $k = \mathbf{y}$, otherwise zero), and $p\left(y|\mathbf{x}\right)$ is the prediction of $\mathbf{x}$. We use two classifiers to simultaneously classify the visual and semantic embeddings to initialize the classifiers, which are used for discrepancy learning later.

**Discrepancy Maximization for Classifiers.** In this step, we freeze the parameters of the encoders ($E^x$, $E^a$) and update the classifiers ($CLS^1$, $CLS^2$) to maximize the discrepancy between the outputs of the two classifiers on the visual and semantic embeddings. Thus, the source and target samples outside the support of task-specific decisions are identified, providing the positive signals to separate the class decision boundaries. Since sliced Wasserstein discrepancy (SWD) [37] provides geometrically meaningful guidance for capturing the natural notion of dissimilarity, we use it to measure the discrepancy between two classifier predictions for visual embeddings ($p_1(y^s|E^x(x^s))$, $p_2(y^s|E^x(x^s))$) and semantic embeddings ($p_1(y^s|E^a(a^s))$, $p_2(y^s|E^a(a^s))$):

$$\mathcal{L}_{DIS^1}(CLS^1, CLS^2) = -\mathcal{L}_{SWD}(E^x(x^s)) - \mathcal{L}_{SWD}(E^a(a^s)), \quad (8)$$

$$\mathcal{L}_{SWD}(\mathbf{x}) = \sum_m \left(\mathcal{R}_{\theta_m} p_1(y|\mathbf{x}), \mathcal{R}_{\theta_m} p_2(y|\mathbf{x})\right), \quad (9)$$

where $\mathcal{R}_{\theta_m}$ is the $m$th one-dimensional linear projection operation, and $\theta$ is a uniform measure on the unit sphere $S^{(d-1)}$ in $\mathbb{R}^d$. The larger the $\mathcal{L}_{DIS}$, the larger the discrepancy.

**Discrepancy Minimization for Encoders.** To encourage the visual embedding ($E^x(x^s)$) and semantic embedding ($E^a(a^s)$) to be aligned well in the structure-aligned space, we freeze the parameters of the two classifiers and update the two encoders ($E^x$, $E^a$) to minimize the discrepancy between the outputs of the two classifiers on $E^x(x^s)$ and $E^a(a^s)$:

$$\mathcal{L}_{DIS^2}(E^x, E^a) = \mathcal{L}_{SWD}(E^x(x^s)) + \mathcal{L}_{SWD}(E^a(a^s)). \tag{10}$$

This encourages the visual and semantic feature manifolds to be closer to each other, and thus the manifold structure variation is circumvented.

## 3.3 Distribution Adaptation

After structure adaptation, HSVA can learn a structure-aligned common space to avoid manifold structure difference between visual and semantic embeddings. However, the structure-aligned common space is still limited by distribution variation. To further learn a distribution-aligned common space for visual and semantic feature representations, we propose distribution adaptation. Existing common space learning methods [8, 9, 7, 10, 11] employ two different mapping functions (encoders) to align the visual and semantic distributions to learn distribution-aligned embedding. In contrast, our main idea is to protect the structure-aligned visual and semantic embeddings in a distribution-aligned space using a common encoder $E^z$, which preserves the structure-aligned representations in the DA. Our DA is first optimized by minimizing the Wasserstein distance between the latent multivariate Gaussian distributions, formulated as follows:

$$\mathcal{L}_{DA}(E^x, E^a, E^z) = \left( \left\| \mu^{x^s} - \mu^{a^s} \right\|_2^2 + \left\| (\delta^{x^s})^{\frac{1}{2}} - (\delta^{a^s})^{\frac{1}{2}} \right\|_F^2 \right)^{\frac{1}{2}}, \tag{11}$$

where $\| \cdot \|_F^2$ denotes the squared matrix Frobenius norm.

In essence, our HSVA is also a generative model, which is different to the GAN/VAE based methods [40, 31, 33] (i.e., learning mechanism and optimization). Due to the bias problem that arises when using generative models for ZSL, the synthesized unseen samples ($E^z(E^a(a^u))$) in the distribution-aligned common space might be unexpectedly close to the seen ones. We propose to explicitly tackle this seen-unseen bias problem by preventing the encoded unseen samples ($E^z(E^a(a^u))$) from colliding with the encoded seen ones ($E^z(E^x(x^s))$)[2]. Since correlation alignment (CORAL) [41] has been shown effective for asymmetric transformations in domain adaptation, we take the CORAL-based metric to measure the discrepancy between seen and unseen class samples. Different from [41], which takes CORAL to decrease the discrepancy between two domains, we aim to increase the discrepancy. Hence, we employ the inverse CORAL (iCORAL), defined upon the numerical negation of CORAL:

$$\mathcal{L}_{iCORAL}(E^x, E^a, E^z) = -CORAL(E^z(E^x(x^s)), E^z(E^a(a^u))). \tag{12}$$

## 3.4 Optimization

Our full model optimizes the partially-aligned variational autoencoders, DA and SA, simultaneously with the following objective function:

$$\mathcal{L}_{HSVA}(E^x, E^a, CLS^1, CLS^2, E^z, D^x, D^a) = \mathcal{L}_{VAE}^x + \mathcal{L}_{VAE}^a + \lambda_1 \mathcal{L}_{REC} + \mathcal{L}_{CLS}$$
$$+ \lambda_2(\mathcal{L}_{DIS^1} + \mathcal{L}_{DIS^2}) + \lambda_3(\mathcal{L}_{iCORAL} + \mathcal{L}_{DA}), \tag{13}$$

where $\lambda_1$, $\lambda_2$, $\lambda_3$ are the weights that control the importance of the related loss terms. Similar to the alternative updating policy for generative adversarial networks (GANs), we alternately train $E^x$, $E^a$, $CLS^1$, $CLS^2$ when optimizing SA. Although, HSVA has three components in total, the whole training process are simultaneous and loss weights of all terms in Eq. 13 are the same for all datasets. The consistently significant results on all datasets show that our model is robust and easy to train.

---

[2]Since $E^z(E^x(x^s))$ and $E^z(E^a(a^u))$ are used to train a ZSL classifier later, we take the visual features of seen classes and semantic features of unseen classes into account.

## 3.5 Classification

Once our HSVA model is learned, the visual and semantic features are encoded as new feature representations in a distribution-aligned common space, where structure variation is aslo considered for classification. Using the reparametrization trick [42], we take $E^x$ and $E^z$ to encode the visual features of seen classes ($x^s$) and unseen classes ($x^u$) into structure- and distribution-aligned feature representations, i.e., $z^{x^s} = E^z(E^x(x^s))$ and $z^{x^u} = E^z(E^x(x^u))$. Analogously, we take $E^a$ and $E^z$ to encode the semantic embeddings $a$ as $z^a = E^z(E^a(a))$, where $z^a = z^{a^s} \cup z^{a^u}$. We employ $z^{x^s}$ ($x^s$ from the training set) and $z^{a^u}$ to train a supervised classifier (e.g., softmax). Once the classifier is trained, we use $z^{x^s}$ ($x^s$ from the test set) and $z^{x^u}$ to test the model. Note that our HSVA is an inductive method as we do not use the visual features of unseen classes for training.

## 4 Experiments

**Datasets.**  We conduct extensive experiments on four well-known ZSL benchmark datasets, including fine-grained datasets (e.g., CUB [43] and SUN [44]) and coarse-grained datasets (e.g., AWA1 [4] and AWA2 [45]). CUB includes 11,788 images of 200 bird classes (seen/unseen classes = 150/50) with 312 attributes. SUN contains 14,340 images from 717 scene classes (seen/unseen classes = 645/72) with 102 attributes. AWA1 and AWA2 contain 30,475 and 37,322 images from 50 animal classes (seen/unseen classes = 40/10) with 85 attributes.

**Implementation Details.**  We use the training splits proposed in [46]. Meanwhile, the visual features are extracted from the 2048-dimensional top-layer pooling units of a CNN backbone (i.e., ResNet-101) pre-trained on ImageNet. The encoders and decoders are multilayer perceptrons (MLPs). We employ the Adam optimizer [47] with $\beta_1 = 0.5$ and $\beta_2 = 0.999$. We use an annealing scheme [48] to increase the weights $\gamma, \lambda_1, \lambda_2, \lambda_3$ with same setting for all datasets. Specifically, $\gamma$ is increased by a rate of 0.0026 per epoch until epoch 90, $\lambda_1$ is increased from epoch 21 to 75 by 0.044 per epoch, and $\lambda_2, \lambda_3$ are increased from epoch 0 to epoch 22 by a rate of 0.54 per epoch. In the CZSL setting, we synthesize 800, 400 and 200 features per unseen class to train the classifier for AWA1, CUB and SUN datasets, respectively. In the GZSL setting, we take 400 synthesized features per unseen class and 200 synthesized features per seen class to train the classifier for all datasets. The dimensions of the structure-aligned and distribution-aligned spaces are set to 2048 and 64 for three datasets (i.e., CUB, AWA1, AWA2), respectively, and 2048 and 128 for SUN benchmark.

**Evaluation Protocols.**  During testing, we follow the unified evaluation protocols proposed in [45]. Specifically, we measure the top-1 accuracy of unseen class ($Acc$) for the CZSL setting. In the GZSL setting, we take the top-1 accuracy on seen classes ($S$) and unseen classes ($U$), as well as their harmonic mean (defined as $H = (2 \times S \times U)/(S + U)$).

| Method | AWA1 $Acc$ | CUB $Acc$ | SUN $Acc$ |
|---|---|---|---|
| DeViSE(NeurIPS'13) [34] | 54.2 | 52.0 | 56.5 |
| DCN(NeurIPS'18) [10] | 65.3 | 56.2 | 61.8 |
| CADA-VAE(CVPR'19) [11] | 63.0 | 59.8 | 61.7 |
| **Our HSVA** | 70.6 | 62.8 | 63.8 |

Table 1: Results of common space learning methods under the CZSL setting on three datasets.

## 4.1 Experimental Results

**Results of Conventional Zero-Shot Learning.**  Table 1 presents the results of CZSL on various datasets. Our HSVA, with a softmax classifier, significantly outperforms other common space learning methods [34, 10, 11] by at least 5.3%, 3.0% and 2.0% on AWA1, CUB and SUN, respectively. Our significant performance improvements demonstrate that hierarchical semantic-visual adaptation can effectively learn an intrinsic common space to represent visual and semantic features. This space is structure- and distribution-aligned and alleviates the gap between heterogeneous feature representations.

| | | AWA1 | | | AWA2 | | | CUB | | | SUN | | |
|---|---|---|---|---|---|---|---|---|---|---|---|---|---|
| | Method | U | S | H | U | S | H | U | S | H | U | S | H |
| Non-common space | CRnet(ICML'19) [49] | 58.1 | 74.7 | 65.4 | - | - | - | 45.5 | 56.8 | 50.5 | 34.1 | 36.5 | 35.3 |
| | PQZSL(CVPR'19) [50] | - | - | - | 31.7 | 70.9 | 43.8 | 43.2 | 51.4 | 46.9 | 35.1 | 35.3 | 35.2 |
| | TCN(ICCV'19) [51] | 49.4 | 76.5 | 60.0 | 61.2 | 65.8 | 63.4 | 52.6 | 52.0 | 52.3 | 31.2 | 37.3 | 34.0 |
| | SGMA(NeurIPS'19) [52] | 37.6 | 87.1 | 52.5 | - | - | - | 36.7 | 71.3 | 48.5 | - | - | - |
| | DVBE(CVPR'20) [53] | - | - | - | 63.6 | 70.8 | 67.0 | 53.2 | 60.2 | 56.5 | 45.0 | 37.2 | 40.7 |
| | DAZLE(CVPR'20) [54] | - | - | - | 60.3 | 75.7 | 67.1 | 59.6 | 56.7 | 58.1 | 52.3 | 24.3 | 33.2 |
| | RGEN(ECCV'20) [55] | - | - | - | 67.1 | 76.5 | 71.5 | 60.0 | 73.5 | 66.1 | 44.0 | 31.7 | 36.8 |
| | Composer(NeurIPS'20) [23] | - | - | - | 62.1 | 77.3 | 68.8 | 63.8 | 56.4 | 59.9 | 55.1 | 22.0 | 31.4 |
| | APN(NeurIPS'20) [25] | - | - | - | 56.5 | 78.0 | 65.5 | 65.3 | 69.3 | 67.2 | 41.9 | 34.0 | 37.6 |
| | f-CLSWGAN(CVPR'18) [46] | 57.9 | 61.4 | 59.6 | - | - | - | 43.7 | 57.7 | 49.7 | 42.6 | 36.6 | 39.4 |
| | f-VAEGAN(CVPR'19) [31] | - | - | - | 57.6 | 70.6 | 63.5 | 48.4 | 60.1 | 53.6 | 45.1 | 38.0 | 41.3 |
| | LisGAN(CVPR'19) [56] | 52.6 | 76.3 | 62.3 | - | - | - | 46.5 | 57.9 | 51.6 | 42.9 | 37.8 | 40.2 |
| | LsrGAN(ECCV'20) [22] | 54.6 | 74.6 | 63.0 | - | - | - | 48.1 | 59.1 | 53.0 | 44.8 | 37.7 | 40.9 |
| | AGZSL(ICLR'21) [26] | - | - | - | 65.1 | 78.9 | 71.3 | 41.4 | 49.7 | 45.2 | 29.9 | 40.2 | 34.3 |
| Common space | DeViSE(NeurIPS'13) [34] | 13.4 | 68.7 | 22.4 | 17.1 | 74.7 | 27.8 | 23.8 | 53.0 | 32.8 | 16.9 | 27.4 | 20.9 |
| | ReViSE(ICCV'17) [9] | 46.1 | 37.1 | 41.1 | 46.4 | 39.7 | 42.8 | 37.6 | 28.3 | 32.3 | 24.3 | 20.1 | 22.0 |
| | DCN(NeurIPS'18) [10] | 25.5 | 84.2 | 39.1 | - | - | - | 28.4 | 60.7 | 38.7 | 25.5 | 37.0 | 30.2 |
| | CADA-VAE(CVPR'19) [11] | 57.3 | 72.8 | 64.1 | 55.8 | 75.0 | 63.9 | 51.6 | 53.5 | 52.4 | 47.2 | 35.7 | 40.6 |
| | SGAL(NeurIPS'19) [19] | 52.7 | 74.0 | 61.5 | 52.5 | 86.3 | 65.3 | 40.9 | 55.3 | 47.0 | 35.5 | 34.4 | 34.9 |
| | **Our HSVA** | 59.3 | 76.6 | 66.8 | 56.7 | 79.8 | 66.3 | 52.7 | 58.3 | 55.3 | 48.6 | 39.0 | 43.3 |

Table 2: State-of-the-art comparison under the GZSL setting on four datasets. The best and second-best results are marked in **red** and **blue**, respectively. The "Common space" and "Non-common space" denote common space learning and non-common space learning methods, respectively. Meanwhile, the non-common space learning methods include the non-generative methods and generative methods.

| | AWA1 | | | AWA2 | | | CUB | | | SUN | | |
|---|---|---|---|---|---|---|---|---|---|---|---|---|
| Method | U | S | H | U | S | H | U | S | H | U | S | H |
| HSVA w/o $\mathcal{L}_{SA}$ | 55.2 | 71.5 | 62.3 | 53.9 | 77.2 | 63.5 | 49.3 | 57.9 | 53.2 | 49.7 | 37.2 | 42.6 |
| HSVA w/o $\mathcal{L}_{DA}$ and $\mathcal{L}_{iCORAL}$ | 31.3 | 70.6 | 43.4 | 26.3 | 78.4 | 39.3 | 41.9 | 56.1 | 47.9 | 42.2 | 35.5 | 38.6 |
| HSVA w/o $\mathcal{L}_{iCORAL}$ | 54.3 | 73.5 | 62.4 | 51.3 | 82.6 | 63.3 | 51.1 | 57.0 | 53.9 | 45.1 | 38.9 | 41.8 |
| HSVA (full) | 59.3 | 76.6 | 66.8 | 56.7 | 79.8 | 66.3 | 52.7 | 58.3 | 55.3 | 48.6 | 39.0 | 43.3 |

Table 3: Ablation study of HSVA under the GZSL setting on four datasets. "HSVA w/o $\mathcal{L}_{SA}$" denotes HSVA without the constraints in SA, "HSVA w/o $\mathcal{L}_{DA}$ and $\mathcal{L}_{iCORAL}$" denotes HSVA without the constraints in DA, "HSVA w/o $\mathcal{L}_{iCORAL}$" denotes HSVA without $\mathcal{L}_{iCORAL}$, and "HSVA (full)" denotes our full model HSVA.

**Results of Generalized Zero-Shot Learning.** Table 2 shows the GZSL performances of various methods, including non-generative methods (direct mapping and model parameter transfer), generative methods, and common space learning methods. Compared to other common space learning models [34, 9, 10, 11, 19], our HSVA achieves significant improvements of at least 2.7%, 1.0%, 2.9% and 2.7% in harmonic mean on AWA1, AWA2, CUB and SUN, respectively. Especially, our HSVA outperforms the latest common space learning method (i.e., SGAL [19]) by a large margin, resulting in harmonic mean improvements of 5.3%, 1.0%, 8.3% and 8.4% on AWA1, AWA2, CUB and SUN, respectively. Since our method can also act as a generative method, we compare it with other state-of-the-art generative methods. The results show that our method performs significantly better on AWA1, CUB and SUN datasets. Our model also achieves competitive results compared to non-generative methods. For example, HSVA performs best results (e.g., the harmonic mean of 43.3) on SUN that is the most challenging benchmark in ZSL. These results consistently demonstrate the superiority and great potential of hierarchical semantic-visual adaptation.

**Ablation Study.** We conduct ablation studies to evaluate the effectiveness of our HSVA in terms of the structure adaptation constraints (denoted as $\mathcal{L}_{SA}$), distribution adaptation constraints (i.e., $\mathcal{L}_{DA} + \mathcal{L}_{iCORAL}$), and only inverse CORAL $\mathcal{L}_{iCORAL}$. Our results are shown in Table 3. HSVA performs poorer than its full model when no constraints are used during structure adaptation, i.e., the harmonic mean drops by 4.5% on AWA1 and 2.1% on CUB. If constraints are not used during distribution adaptation, HSVA achieves very poor results compared to its full model, i.e., the harmonic mean drops by more than 23.4% on coarse-grained datasets and 4.7% on fine-grained datasets. These results show that distribution alignment is essential for common space learning, while structure adaptation is necessary for semantic-visual adaptation. This is typically neglected by distribution alignment methods with one-step adaptation [8, 9, 7, 10, 11, 19]. Moreover, $\mathcal{L}_{iCORAL}$ pushes

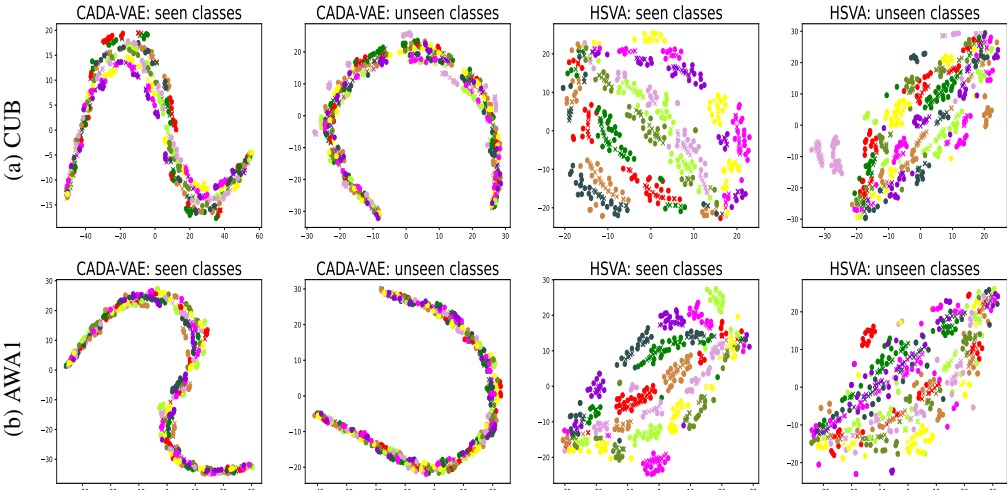

Figure 3: t-SNE visualization [14] of visual/semantic features learned by our HSVA and CADA-VAE [11] for the same seen and unseen classes. Different colors denote different classes. The "o" and "x" indicate visual and semantic features, respectively. We conduct experiments on 10 classes of (a) CUB and (b) AWA1. CADA-VAE clearly maps the visual and semantic features into various manifolds focusing on distribution alignment, while HSVA learns an intrinsic common space for discriminative feature representations by adopting sequential structure and distribution adaptation.

unseen classes away from seen classes, which explicitly tackles the seen-unseen bias problem. HSVA cooperates with iCORAL to improve the accuracy of classification using softmax, achieving harmonic mean improvement of 4.4%, 3.0%, 1.4%, and 1.5% on AWA1, AWA2, CUB, and SUN, respectively.

**Qualitative Results.** We first qualitatively investigate the difference between our HSVA and one-step adaptation methods (e.g., CADA-VAE [11]) for common space learning. We conduct t-SNE visualization [14] of visual/semantic features mapped by our HSVA and CADA-VAE on 10 classes from (a) CUB and (b) AWA1. As shown in Figure 3, CADA-VAE maps semantic and visual features into different manifolds, which confuses the visual and semantic features of various categories, resulting in misclassification for some samples. This intuitively shows that the existing common space learning methods neglect the different structures of the heterogeneous feature representations in the visual and semantic domains. In contrast, HSVA learns an intrinsic common space that better bridges the gap between the heterogeneous feature representations for effective knowledge transfer, using structure and distribution adaptation. Thus, hierarchical semantic-visual adaptation is an effective solution for common space learning in ZSL.

In addition, unlike other methods [8, 9, 7, 10, 11] that can only align visual and semantic distributions, our HSVA can take class relationships into account to learn discriminatively aligned feature representations. As shown in Figure 3, CADA-VAE maps visual and semantic features into compact manifolds, resulting in confused class relationships. In contrast, HSVA learns discriminative structure- and distribution-aligned features for the visual and semantic feature representations, which enables HSVA to achieve better ZSL classification.

## 5 Conclusion

In this work, we develop a zero-shot learning framework, i.e. hierarchical semantic-visual adaptation (HSVA), to learn an intrinsic common space for visual and semantic feature representations. By conducting structure and distribution adaptation using two partially-aligned variational autoencoders, our model effectively tackles the problem of heterogeneous feature representations and bridges the gap between the visual and semantic domains. We demonstrate that HSVA achieves consistent improvement over the current state-of-the-art methods on four ZSL benchmarks. We qualitatively verify that our hierarchical semantic-visual adaptation framework can align the visual and semantic domains in terms of both structure and distribution, while other common space learning methods only

take distribution alignment into account. This intuitively shows that the learned feature representations of HSVA are more discriminative and accurate than other methods. To the best of our knowledge, we are the first to investigate the different structural characteristics of the visual and semantic domains in ZSL. We believe that our work will facilitate the development of stronger visual-and-language learning systems, including zero-shot learning, visual question answering [57], image captioning [58], natural language for visual reasoning [59], heterogeneous domain adaptation [60].

## Acknowledgements

This work is partially supported by NSFC (61772220), Special projects for technological innovation in Hubei Province (2018ACA135) and Key R&D Plan of Hubei Province (2020BAB027).

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
