# 1 Appendix

In this appendix, Section A summarizes the architecture details of HSVA. Section B provides the key hyperparameter analyses and setting in our experiments.

## A  Network Topology

Our proposed HSVA consists of two partially-aligned variational autoencoders, which include three encoders (i.e., $E^x$, $E^a$, and $E^z$), and two decoders (i.e., $D^x$, and $D^a$). As described in the paper that $E^x$, $E^a$, $E^z$, $D^x$, and $D^a$ are MLP architectures, we present the architecture details of them as shown in Table 3.

| | |
|---|---|
| Visual encoder ($E^x$) | **Input**: $x$, size=2048;
**hidden layer**: Fully connected, neurons=4096; LeakyReLU;
**Output**: Fully connected, neurons=2048; |
| Semantic encoder ($E^a$) | **Input**: $x$, size=$\lvert Att\rvert$;
**hidden layer**: Fully connected, neurons=4096; LeakyReLU;
**Output**: Fully connected, neurons=2048; |
| Classifiers ($CLS^1$/$CLS^2$) | **Input**: $E^x(x)$ or $E^a(a)$, size =2048;
**hidden layer**: Fully connected, neurons=512; BachNorm, LeakyReLU;
**Output**: Fully connected, neurons=$\lvert Seen\rvert$; |
| Common encoder ($E^z$) | **Input**: $E^x(x)$ or $E^a(a)$, size=2048;
**hidden layer**: Fully connected, neurons=2048; LeakyReLU;
**hidden layer**: Fully connected, neurons=64*2; LeakyReLU;
**encoding layer**: $\mu^x$=64 and $\delta^x$=64, or $\mu^a$=64 and $\delta^a$=64;
**Output**: Reparametrization, $z^x$ or $z^a$, neurons=64; |
| Visual decoder ($D^x$) | **Input**: $z^x$, size=64;
**hidden layer**: Fully connected, neurons=4096; LeakyReLU;
**Output**: Fully connected, neurons=2048; |
| Semantic decoder ($D^a$) | **Input**: $z^a$, size=64;
**hidden layer**: Fully connected, neurons=4096; LeakyReLU;
**Output**: Fully connected, neurons=$\lvert Att\rvert$; |

Table 3:  Network topology of HSVA. $\lvert Att\rvert$ is the dimensionality of semantic vectors per class, e.g., $\lvert Att\rvert = 312$ in CUB. $\lvert Seen\rvert$ denotes the numbers of seen classes, e.g., $\lvert Seen\rvert = 150$ in CUB.

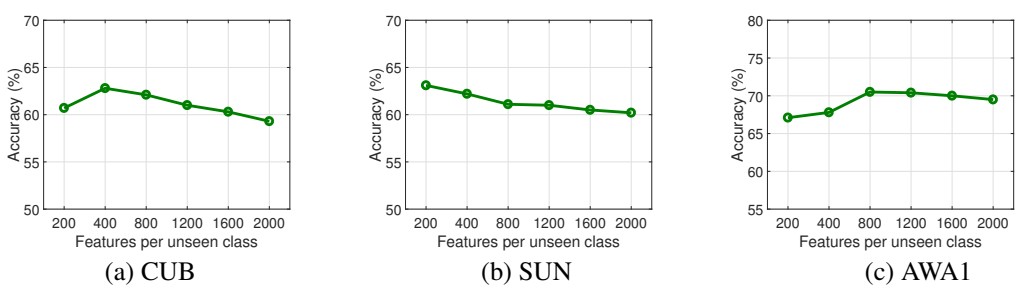

(a) CUB          (b) SUN          (c) AWA1

Figure 4: Evaluating the effect of the number of synthesized latent features per unseen class on (a) CUB, (b) SUN and (c) AWA1 in CZSL setting.

## B  Hyperparameter Analysis

**Features of Per Unseen Class in CZSL Setting** ($N_u$)**.** We evaluate the effect of the number of latent features per unseen class in CZSL. Since we only need to synthesize unseen features of unseen classes for training a classifier, We try a wide range of $N_u$ (i.e., $N_u = \{200, 400, 800, 1200, 1600, 2000\}$) for evaluation on CUB, SUN and AWA1 datasetsas shown in Figure 4. Overall, the performance of HSVA is insensitive to the number of latent features per unseen class in CZSL. Targeting on better results, we set $N_u$ as 400, 200 and 800 for CUB, SUN and AWA1, respectively.

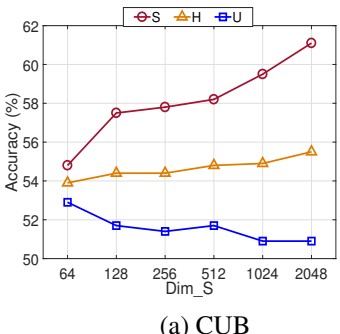
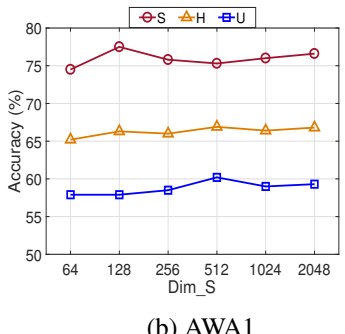

(a) CUB                (b) AWA1

Figure 5: The influence of the dimentionality of the latent features in the structure-aligned common space ($Dim\_S$).

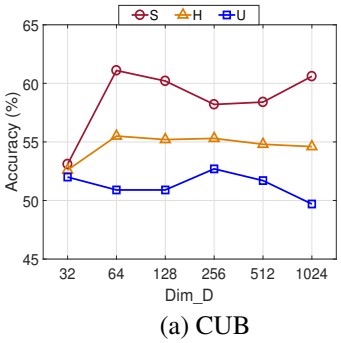
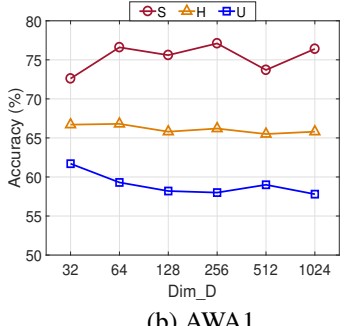

(a) CUB                (b) AWA1

Figure 6: The influence of the dimentionality of the latent features in the distribution-aligned common space ($Dim\_D$).

**Features of Per Seen and Unseen Class in GZSL Setting ($N_s$ and $N_u$).** We analyze the effect of the number of latent features per class in GZSL. We try a wide range of $N_s$ and $N_u$ (i.e., $N_s = \{100, 200, 400\}$ and $N_u = \{100, 200, 400, 800, 1200\}$) for evaluation on CUB and AWA1 datasets, resuting in a total of 15 pairs of ($N_s$, $N_u$), as shown in Figure 7. Since the visual features possesses more discriminative information, we should set $N_u$ larger than $N_s$. Compared to HSVA using $N_s/N_u = 1/1$, HSVA improves classification accuracy using $N_s/N_u = 1/2$, achieving top-1 accuracy on unseen classes (Harmonic mean) improvement at least 17.5%(9.5%) and 36.3%(30.5%) on fine-grined dataset (e.g., CUB) and coarse-grained dataset (AWA1), respectively. Note that HSVA achieves better results on seen classes when $N_s/N_u$ is set to larger than $1/2$. To trade-off top-1 accuracy on seen and unseen, we set ($N_s$, $N_u$) = (200, 400) to conduct all experiments.

**Dimensionality of Latent Features in Structure- and Distribution-Aligned Common Space.** Here we show how to set the dimensionality of the latent features in structure- and distribution-aligned common space, denoted as $Dim\_S$ and $Dim\_D$, respectively. As shown in Figure 5 and Figure 6, HSVA perform steadily on the coarse-grained dataset (e.g., AWA1) while it is sensitive to $Dim\_S$ and $Dim\_D$ on fine-grained datasets (e.g., CUB). On the fine-grained datasets, HSVA increases its accuracy on seen classes and decreases its accuracy on unseen classes when $Dim\_S$ and $Dim\_D$ are increased. We note that HSVA achieves significant results when $Dim\_S = 2048$ and $Dim\_D = 64$, thus we set $Dim\_S$ and $Dim\_D$ to 2048 and 64 respectively on CUB and AWA1.

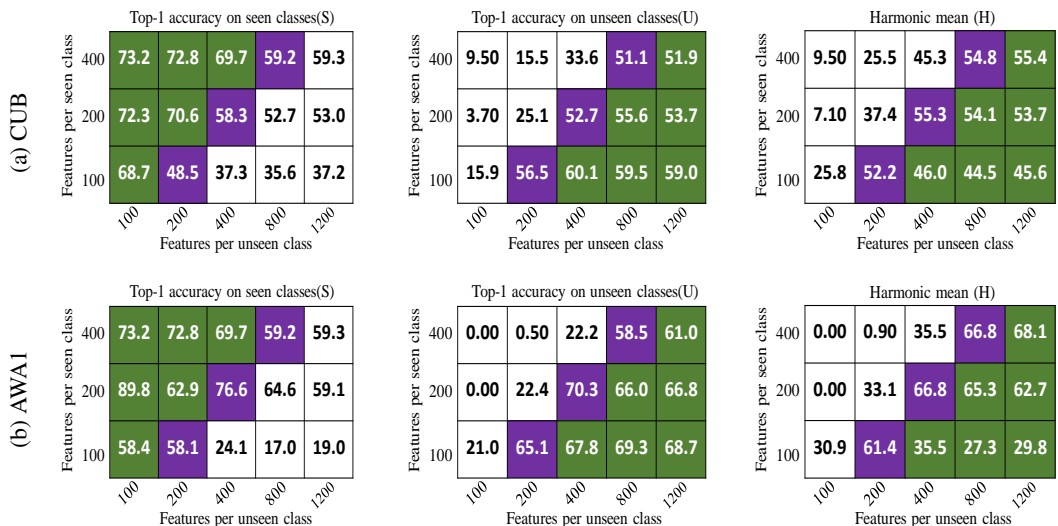

Figure 7: Evaluating the effect of the number of synthesized latent features per seen/unseen class on (a) CUB and (b) AWA1 in GZSL setting.