# OpenReview forum: "HSVA: Hierarchical Semantic-Visual Adaptation for Zero-Shot Learning"
_NeurIPS.cc/2021/Conference — NeurIPS 2021 Poster_

### Official Review · Reviewer_Nr4Y · 2021-07-08

**Rating:** 7
**Confidence:** 5

**Summary:**

This paper proposes an hierarchical semantic-visual adaptation framework to migrate the distribution and structure disagreement in generative zero-shot learning (ZSL) models. This network adopts hierarchical two-step adaptation. The task-specific adaptation is complemented by training two task-specific encoders with a supervised adversarial discrepancy module. The distribution adaptation is achieved by minimizing the Wasserstein distance between visual and semantic distributions. Quantitative results on four widely used benchmarks indicates the effectiveness of the proposed method. Qualitative visualization shows the network learns intrinsic common space that better bridges the gap between the heterogeneous feature representations.


**Ethical Concerns:**

There is no negative societal impact of this work.


**Limitations And Societal Impact:**

This paper deals with an interesting problem but some concerns are unsolved and make the contribution somehow weak (see detailed comments as before). I would like to raise my rating if the author’s feedback properly addresses the concerns.

Following are minor questions:

Line 113: “ $a^s$ are learned by a language model and …” is not accurate. In this paper, $a^s$ is collected from human annotation not learned from the language model.



**Main Review:**

[Originality] The proposed two-step adaptation is novel and seems interesting. The model deals with the “structure variation” problem in one-step adaptation. However, the related work part misses comparison with two important GAN and VAE based related work [1r,25]. The author is encouraged to fully discuss the distinction of the proposed method and state out the novelty.

[Clarity] The paper is very well organized and easy to follow. However, the definition of “distribution” and “structure” variation between semantic and visual features is not well explained, which hinders the motivation of this paper.
For instance, in Figure 1 the author argues that one-step adaptation causes the semantic and visual distributions to be located in different manifolds, however, it seems like the two distributions are located on the same manifold from the t-SNE visualization in (c). And the main problem in (c) seems like high inner-class variance and low inter-class variance.

[Quality] The effectiveness of the proposed components is verified by ablation study and qualitative results. However, I have one concern over the architecture and the experiment design the “structure adaptation” section.
In the structure adaptation section, two “task-specific” classifiers are proposed to classify feature and semantic features into the corresponding category and a SAD module is proposed to minimize the discrepancy between the classifiers’ outputs. However, there lack of explanation for this specific design since SA is the most distinct part of this paper compared to other models [29,30,11].
1. What structure do the two classifiers represent? Why choose two classifiers, not three or five?
2. It seems like the input and constraints of the two classifiers are exactly the same. If the two classifiers are initialized with the same parameter, will they turn out to be the same? Then why would we need two classifiers? Will one classifier also work? Several ablations for the SA section are missing. For instance, what is the performance of (1) using two classifiers without $L_{DIS}$, (2) using one or three classifiers?

[Significance]
Quantitative results on four widely used benchmarks indicate the effectiveness of the proposed method. However, there are two concerns towards the qualitative results shown in Figure 3.
Showing qualitative ablation on the proposed loss functions will shed more insights on the contribution of this work, e.g., what’s the effect of removing SA or DA on Figure 3?
How does the author choose the 10 classes in Figure 3? It seems that CADA-VAE is much worse than the proposed HSVA qualitatively. However, from quantitative results, CADA-VAE is achieving comparable results (only 2-3% lower) with the proposed method. Why is this happening? Does it mean that the quality of feature distribution is not the main factor of high ZSL accuracy?

[1r] Narayan, Sanath, et al. "Latent embedding feedback and discriminative features for zero-shot classification." ECCV, 2020.


[Post-rebuttal]  The authors have addressed my concerns. After reading other reviewers' comments, I agree to accept the paper as it is interesting and sound.

**Time Spent Reviewing:**

5

---

> ### Author Response · Authors · 2021-08-08
> **Authors' response to Reviewer Nr4Y**
>
> Thank you for the comprehensive reviews and detailed comments! We are very happy to help to address your concerns!
>
>
> **Q1:** Originality.
>
> **A1:** We are very pleased that you approve the novelty and interesting aspect of our idea. In essence, our method is also a generative model. However, there are two differences between our method with the GAN/VAE based methods [1r,25]:
>
> 1) The GAN/VAE based generative methods [1r,25] aim to learn a generator to present the semantic-visual mapping, while our HSVA attempts to jointly map the visual and semantic into an intrinsic common space using the visual encoder ($E^x$), the semantic encoder ($E^a$) and common encoder ($E^z$).
>
> 2) The discriminator of GAN-based generative methods only tries to distinguish the features as real or fake and thus does not consider task-specific decision boundaries among different classes. By contrast, our HSVA attempts to align the semantic and visual domains by optimizing the discrepancy between two classifiers' outputs.
>
> We will add these discussions in the revision.
>
> *[1r] Narayan, Sanath, et al. "Latent embedding feedback and discriminative features for zero-shot classification." ECCV, 2020.*
>
> **Q2:** Clarity.
>
> **A2:** It’s our pleasure to make you feel that our paper is easy to follow. The  “distribution” represents the feature distribution, and the “structure” represents the manifold structure of the feature space. As shown in Fig. 1(a)，the visual and semantic features are nearly aligned in two manifolds since the one-step adaptation methods haven’t taken the structure variation into account. In Fig. 1(c), it seems like the two distributions are located on the same manifold by the 2D t-SNE visualization. This is because: 1) The 2D t-SNE projection of high-dimensional data cannot absolutely capture the whole properties in this high-dimensional feature space; 2) The visual and semantic features of CADA-VAE are usually aligned nearly in two individual manifolds by solely relying on the Euclidean distance during training, and, sometimes, some mapped samples from visual and/or semantic domains are even located in the opposing domain’s manifolds (see amended Fig. 1(a)); 3) CADA-VAE training haven’t considered the relationships among different classes, which inevitably leads to a confusion subspace. Thus, the 2D feature visualization in Fig 1(c), more or less, shall show us a “high inner-class variance and low inter-class variance’’ (manifold confusing) layout.
>
> By contrast, our HSVA still keeps good feature representations in the 2D space as shown in Fig. 1(d). This shows the superiority of our method.
>
>
> **Q3:** Quality.
>
> **A3:** In the structure adaptation procession, we take the SAD module to align the visual and semantic domains by utilizing the task-specific decision boundaries. This is different from the one-step adaptation methods, which aim to completely match the feature distributions between different domains.  As such, we can align the manifold structures of visual and semantic domains. Specifically, we propose to optimize the discrepancy between two classifiers’ outputs to simultaneously detect semantic features that are far from the support of the visual features and visual features that are far from the support of the semantic features.
> To represent the discrepancy effectively, we take two classifiers. We conclude that 1) If one classifier is used, the discrepancy cannot be distinctively represented. 2) if more classifiers are used, i.e., three classifiers, the adversarial optimization is hard to be performed. Meanwhile, we experimentally show that two classifiers are good for SAD optimization. Besides the ablation in Table 3, we also conduct ablation studies to confirm our claims below.
>
> * Results of using two classifiers without $L_{DIS}$ in SA for HSVA.
>
> | Dataset | U | S | H |
> | :-----: | :-----: | :-----: | :-----: |
> | AWA1 | 54.6 | 77.2 | 64.0|
> | AWA2 | 53.1 | 80.3 | 63.9 |
> | CUB   | 50.7 |57.3     | 53.8 |
> | SUN   | 45.5 |     39.1 | 42.1 |
>
> * Results of using one classifier for representing the discrepancy in SA for HSVA.
>
> | Dataset | U | S | H |
> | :-----: | :-----: | :-----: | :-----: |
> | AWA1 | 57.0 | 74.2 | 64.5 |
> | AWA2 | 55.3 | 77.2 | 64.4 |
> | CUB   | 50.7 | 58.7 | 54.4 |
> | SUN   | 48.7 | 37.7 | 42.5 |
>
> * Results of using two classifiers for representing the discrepancy in SA for HSVA.
>
> | Dataset | U | S | H |
> | :-----: | :-----: | :-----: | :-----: |
> AWA1	| 59.3 | 76.6 |66.8 |
> AWA2	| 56.7 | 79.8 | 66.3 |
> CUB	| 52.7 |  58.3 | 55.3 |
> SUN	| 48.6 |  39.0 | 	43.3 |
>
>
>
> * Results of using three classifiers for representing the discrepancy in SA for HSVA.
>
> | Dataset | U | S | H |
> | :-----: | :-----: | :-----: | :-----: |
> | AWA1 | 55.6 | 76.1 | 64.3 |
> | AWA2 | 55.1 | 81.0 | 63.0 |
> | CUB   | 50.3 | 58.7    | 54.2 |
> | SUN   | 45.6 |38.8     | 41.9 |
>
>
> **Q4:** Significance.
>
> **A4:** To intuitively show the effectiveness of our method, we take the t-SNE visualization of visual/semantic features for 10 seen classes and 10 unseen classes in Fig. 3. Note that we select the first 10 classes for visualization according to the data loader. As shown in Fig. 3, CADA-VAE maps semantic and visual features into different manifolds as it neglects the different structures of the heterogeneous feature representations in the visual and semantic domains, which confuses the visual and semantic features of various categories heavily. As a result, the upper bound of the recognition performance of the existing common space learning methods for ZSL is potentially limited. In contrast, our HSVA learns an intrinsic common space that better bridges the gap between the heterogeneous feature representations for effective knowledge transfer, using both structure and distribution adaptation. As such, HSVA achieves an improvement of harmonic mean by 2-3\% on all datasets, which shows the effectiveness of our method. In the ZSL community, ~2-3% improvement is already very difficult for those popular datasets. In addition, HSVA achieves state-of-the-art results on the most challenging dataset (SUN). Although HSVA is much better than CADA-VAE qualitatively, the features from some tough classes learned by HSVA are still confusing to some extent. This indicates that HSVA has not tackled the problem of heterogeneous features alignment thoroughly. However, HSVA provides a novel insight for facilitating the development of stronger visual-and-language learning systems, including zero-shot learning, visual question answering [52], natural language for visual reasoning [53], etc.
>
> To shed more insights on the contributions of this work, we further conduct qualitative ablation on the proposed loss functions (i.e., HSVA w/o $L_{DA}$ , HSVA w/o $L_{SA}$). Note that the visualized classes are the same as Fig. 3 in the paper. These results are presented at the anonymous project page [https://github.com/anonymou-ssubmition/HSVA-NeurIPS-21](https://github.com/anonymou-ssubmition/HSVA-NeurIPS-21). The diversity of features synthesized by the semantic vector is extremely limited when HSVA without $L_{DA}$, and thus the model is overfitting to seen classes heavily. This decreases the classification accuracy of unseen classes obviously, e.g., dropping down 28.0\%, 30.4\% and 10.8\% on AWA1, AWA2 and CUB, respectively. For HSVA without $L_{SA}$ constraint, it will be limited to structure variation and thus the classification accuracies are also decreased. Thus, we should simultaneously take the structure and distribution adaptation to learn an intrinsic common space for representing the features of visual and semantic.
>
>
> **Q5:** Line 113: “ $a^s$ are learned by a language model and …” is not accurate.
>
> **A5:** We will correct it in the revision.

---

> ### Author Response · Authors · 2021-08-24
> **Looking Forward to the Comments of  Reviewer Nr4Y After Rebuttal**
>
> Dear Reviewer Nr4Y,
>
> We have properly addressed all your concerns and provided explanations to all questions. If there are still unclear parts to you, please kindly let us know. We are very glad to further discuss them.
>
> Regards,
> Authors

---

### Official Review · Reviewer_JPqU · 2021-07-15

**Rating:** 6
**Confidence:** 3

**Summary:**

In zero-shot learning (ZSL) problem, feature representations of distinct semantic and visual domains are intrinsically heterogeneous, thus having both distribution and structure variations. To this problem, this paper proposes a novel HSVA framework that aligns the semantic and visual domains by structure adaptation (SA) and distribution adaptation (DA). SA, consisting of two encoders, aims to encode the visual data and semantic data into a structure-aligned common space, by using a SAD module. Then, DA further explores a common encoder to align the visual and semantic distributions. This paper also conducts extensive experiments to show the effectiveness of the proposed framework for both conventional ZSL and generalized ZSL settings.

**Main Review:**

Strength:
- Significance and novelty: The main idea of this paper is to learn an intrinsic common space in which the visual and semantic features can be aligned from the perspective of both structure and distribution. The idea of two-step semantic-visual adaptation seems interesting and potentially inspirational for other ZSL works.
- The authors conducted extensive experiments on benchmarks to show its effectiveness.
- The t-SNE visualization of visual/semantic features in Figure 3 intuitively show the manifold of the feature space.

Weakness:
- Model complexity: Though the idea seems novel, the model is somewhat cumbersome to me, e.g. the objective function has 8 terms and 4 hyper-parameters (include $\gamma$). It may be very challenging to find the optimal hyper-parameters for each specific dataset.
- Data sets and training: In addition, in benchmarks like AwA, each class has a unique attribute vector and multiple training samples. Whether the imbalance between visual domain and semantic domain affects the training process also requires discussion.
- Experiments and performance: From the experimental results in Table 2, it can be seen that the proposed method has no advantage in performance comparison on the fine-grained data set, i.e. CUB. Some analysis of the above phenomena may contribute to a better understanding of the proposed method.

**Time Spent Reviewing:**

20

---

> ### Author Response · Authors · 2021-08-08
> **Authors' response to Reviewer JPqU**
>
> Thank you for the comprehensive reviews and constructive comments! We are very happy to help to address your concerns!
>
> **Q1:** Model complexity.
>
> **A1:** Although our total loss function has 8 terms and 4 hyper-parameters, the definition and usage of these losses have been clearly introduced. Meanwhile, several losses are formulated with similar forms, which are cooperated for easy optimization, e.g., $L_{VAE}^x$ and $L_{VAE}^a$ , $L_{DIS^1}$ and $L_{DIS^2}$ .  Additionally, we use the same loss weight settings for all datasets. The consistently significant results on all datasets show that our model is robust and easy to train. Furthermore, the ablation studies intuitively show the effectiveness of these loss functions.
>
> **Q2:** Dataset and training.
>
> **A2:** This is a constructive comment. Indeed, there is an imbalance between visual and semantic domains as each class has a unique attribute vector and multiple training samples. This leads the model to be easily overfitting to seen classes since the unseen features are generated with semantic vector used for training classifier, known as the bias problem. As such, we take two solutions to tackle this challenge: 1) We employ the inverse CORAL (iCORAL) to push the synthesized unseen features away from seen features during training HSVA. 2) We generate more features per unseen class ($N_u$) based on the semantic vector than the features of per seen class ($N_s$) based on visual samples during training the classifier. As shown in Fig. 7 (see Appendix), compared to HSVA using $N_s/ N_u= 1/1$, HSVA improves classification accuracy using $N_s/N_u=1/2$, achieving top-1 accuracy on unseen classes (Harmonic mean) improvement at least 17.5%(9.5%) and 36.3%(30.5%) on fine-grained dataset (e.g., CUB) and coarse-grained dataset (AWA1), respectively. We will add these discussions in the revision.
>
> **Q3:** Experiments and performance.
>
> **A3:** We should emphasize that our method performs well on all benchmarks. For example, our HSVA outperforms the latest common space learning method (i.e., SGAL [17]) by a large margin, resulting in harmonic mean improvements of 5.3%, 1.0%, 8.3% and 8.4% on AWA1, AWA2, CUB and SUN, respectively.
> From Table 2, it seems that RGEN [47] and APN [49] perform better than our HSVA on AWA2 and CUB. The reason is that RGEN and APN are the end-to-end models for fine-tuning the visual features, and thus the cross-dataset bias between ImageNet and ZSL benchmarks is alleviated. Xian [25] conducted experiments to show that fine-tuning can effectively improve the performance of ZSL (e.g., improvement of harmonic mean by 15.3% on CUB). Furthermore, our HSVA achieves great improvements of harmonic mean by 6.5% and 5.7% over RGEN and APN on the most challenging benchmark (i.e., SUN), respectively. This further shows the effectiveness of our method. We will further clarify these to avoid confusion in the revision.
>
> **We are pleased that you find our idea interesting. we hope you feel the flaws are addressed in our response.**

---

> ### Author Response · Authors · 2021-08-24
> **Looking Forward to the Comments of  Reviewer JPqU After Rebuttal**
>
> Dear Reviewer JPqU,
>
> We have tried our best to address all the concerns and provided explanations to all questions. If there are still unclear parts to you, please kindly let us know. We are very glad to further discuss them.
>
> Regards,
> Authors

---

### Official Review · Reviewer_TyLe · 2021-07-15

**Rating:** 6
**Confidence:** 4

**Summary:**

This paper proposes a new method for zero-shot learning. It enforces the classes are well aligned by using a domain-specific discrepancy on an intermediate space of the encoder. The method is evaluated on benchmark datasets and achieves good results.

**Limitations And Societal Impact:**

The paper didn't discuss the limitation or potential negative social impact.

**Main Review:**

This paper proposes a novel method for ZSL. It integrates a two-step encoding process to align the representations from visual and semantic spaces. Compared to previous methods in the same category, this method has an additional encoder that maps the visual features and semantic attributes to a common space. In this common space, it leverages the sliced Wasserstein discrepancy to learn the decision boundary for each domain, such that the encoding can be better aligned. In the empirical evaluation, the proposed method shows superior performance to the corresponding one-step methods.

The originality of this paper is kind of weak to me because the proposed method is a combination of existing works. It integrates the sliced Wasserstein discrepancy into the CADA-VAE framework by introducing an additional encoding layer. It also integrates CORAL to improve the class-specific discrepancy in the distribution alignment stage. These integrations are rather empirical. It is good to see composing these ideas together can improve the CADA-VAE framework, but it makes the contribution of this paper less significant.

Moreover, the proposed method has large performance gaps between state-of-the-art methods on popular benchmark datasets like AWA2 and CUB. This also makes an empirical method less significant.

The writing of this paper is not very good, and it is hard to follow for some parts.

-----------------
Post-rebuttal
After reading the authors' response and other reviewers' comments, I agree that the proposed method in this paper is interesting and more than just a stack of existing works. And it is also different from previous works. So I raise my score.

**Time Spent Reviewing:**

5

---

> ### Author Response · Authors · 2021-08-08
> **Authors' response to Reviewer TyLe**
>
> Thanks for your precious time to review our submission and provide these comments!  Here are our responses to your concerns:
>
> **Q1:** The originality of this paper.
>
> **A1:** We have to first emphasize the main contribution of this paper: 1) To the best of our knowledge, we are the first to raise the heterogeneous nature of the feature representations in the visual and semantic domains in the ZSL field, which intrinsically contain both distribution and structure variations. We intuitively and thoroughly analyze and model this important yet long-time neglected insight and propose HSVA. 2) HSVA aims to align the semantic and visual domains by adopting a hierarchical two-step adaptation (i.e., structure adaptation and distribution adaptation) under a unified partially-aligned variational autoencoder framework, which is a novel framework for common space learning in ZSL. 3) The extensive experiments demonstrate the effectiveness of our method.
>
> Furthermore, we have to emphasize that our HSVA is not a simple stacked tool.
>
> 1) Difference to sliced Wasserstein discrepancy (SWD): Different from sliced Wasserstein discrepancy, which only detects the target samples (e.g., semantic embeddings $E^a(a)$) to learn a single encoder (generator), SAD simultaneously detects the source samples (i.e., visual embeddings $E^x(x)$) and target samples (i.e., semantic embeddings $E^a(a)$) to learn a good structure-aligned common space with two task-specific encoders (stated in L154-157). We argue that bidirectional learning can better align the structure of visual and semantic domains. To show the advantages of SAD over SWD，we conduct an experiment that we take SWD for optimizing the SA module. Results on various datasets are shown below. These results clearly show the superior of SAD used for optimizing HSVA.
>
> * Results of using  SAD for optimizing SA of HSVA.
>
> | Dataset | U | S | H |
> | :-----: | :-----: | :-----: | :-----: |
> | AWA1 | 59.3 | 76.6 | 66.8 |
> | AWA2 |56.7  |79.8 | 66.3 |
> | CUB  | 52.7 | 58.3 | 55.3 |
> | SUN  | 48.6 | 39.0 | 43.3 |
>
> * Results of replacing SAD with SWD for optimizing SA of HSVA.
>
> | Dataset | U | S | H |
> | :-----: | :-----: | :-----: | :-----: |
> | AWA1 | 54.9 | 78.3 | 64.6 |
> | AWA2 | 53.8 | 79.6 | 64.2 |
> | CUB  | 49.8 | 58.5 | 53.8 |
> | SUN  | 46.7 | 38.5 | 42.2 |
>
>
>
> 2) Difference to CADA-VAE: CADA-VAE aligns the feature distributions of semantic and visual domains extracted from two different encoders, while our HSVA aligns the feature distributions of semantic and visual domains learned from a common encoder. Furthermore, our HSVA also attempts to reduce the bias problem in DA, which is neglected by CADA-VAE.
>
> 3) Difference to CORAL: CORAL is originally used to align two distributions in domain adaptation by decreasing the discrepancy between two domains, on the contrary, we employ inverse CORAL (iCORAL) to increase the discrepancy between seen and unseen class features.
>
> **Q2:** Performance gaps between our methods and state-of-the-art methods on popular AWA2 and CUB.
>
> **A2:**  We should emphasize that our method performs well on all benchmarks. For example, our HSVA outperforms the latest common space learning method (i.e., SGAL [17]) by a large margin, resulting in harmonic mean improvements of 5.3%, 1.0%, 8.3% and 8.4% on AWA1, AWA2, CUB and SUN, respectively.
> From Table 2, it seems that RGEN [47] and APN [49] perform better than our HSVA on AWA2 and CUB. The reason is that RGEN and APN are end-to-end models for fine-tuning the visual features, and thus the cross-dataset bias between ImageNet and ZSL benchmarks is alleviated. Xian [25] conducted experiments to show that fine-tuning can effectively improve the performance of ZSL (e.g., improvement of harmonic mean by 15.3% on CUB). Furthermore, our HSVA achieves great improvements of harmonic mean by 6.5% and 5.7% over RGEN and APN on the most challenging benchmark (i.e., SUN), respectively. This further shows the effectiveness of our method. We will further clarify these to avoid confusion in the revision.

---

### Official Review · Reviewer_jKPX · 2021-07-16

**Rating:** 7
**Confidence:** 4

**Summary:**

The paper proposes a new two-step, hierarchical adaptation approach for Zero-Shot Learning. The first step is structure adaptation where visual data and target/semantic data (eg attributes) are encoded into an aligned common space. This is similar to prior work, but is accomplished using a new module that adversarially minimizes the discrepancy between the visual and semantic-based classifiers. The second step is distribution adaptation which seeks to align the visual and semantic feature distributions using a common encoder.

The paper ablates the contribution of both structure and distribution alignment on several ZSL benchmarks, showing significant improvements. It also compares favorably against all prior common space ZSL methods on all benchmarks, and against all prior work (common and non-common space) on several benchmarks.

**Limitations And Societal Impact:**

This paper aims to improve zero-shot learning (an established research area). As such, it does not introduce new negative societal impacts.

**Main Review:**

My main concern is with the complexity of the approach, and the lack of ablation studies to identify which components are most important.  However, the final loss function is quite complicated. Although all components seem to be conceptually sound, given the complexity of the overall method (and the introduction of several modifications to prior work), I believe the authors should have conducted more detailed ablation studies. For example, Ln 146 mentions that prior work uses MMD for structure alignment, but this paper experiments with a new module SAD. This change should have been empirically validated. Does the adversarial approach improve generalization?

I hope the authors can provide more insights into additional components of the proposed method, as I will reassess my score based on it.

-------

Post rebuttal:

After carefully considering the comments of all reviewers, as well as the rebuttal, I'm still leaning towards acceptance.
The paper introduces an interesting and well-motivated development over VAE-style approaches for ZSL, and achieves good results on standard ZSL benchmarks.
My main concern with respect to this paper was the lack of ablations, given the complexity of the training schema. I appreciate that the authors try to address this concern in rebuttal. The provided ablations make sense and cast light on the advantages of different components of the method. I strongly encourage the authors to add these ablations to the paper.

**Time Spent Reviewing:**

4 hours split over 2 days

---

> ### Author Response · Authors · 2021-08-06
> **Authors' response to Reviewer jKPX**
>
> Thank you for reviewing our submission and the comprehensive comments! We are very happy to answer the two aspects with regard to the complexity of our method and conducting more ablation studies.
>
> **Q1:** The complexity of our method.
>
> **A1:**  Our HSVA has a clear intuition of leveraging sequential structure and distribution adaptation for advancing ZSL, under a unified partially-aligned variational autoencoder framework. This two-step VAE-style scenario thus inevitably leads to the first three terms in Eq. (13) for preserving the merits of VAE, whose function has been clearly depicted in Sec. 3.1. The training of these VAE-related terms, in essense, is simple and can follow the established paradigm [11]. For the 4-6 terms in Eq. (13), as in Sec. 3.2, their functions are for the structure adaptation, which utilizes the adversarial training strategy among two classifiers and the encoders (i.e., $E^x(x), E^a(a)$). Finally, the last two terms in Eq. (13) aim to conduct distribution adaptation. Although, HSVA has three components in total, the whole training process are simultaneous and loss weights of all terms in Eq. (13) are the same for all datasets. The consistently significant results on all datasets show that our model is robust and easy to train. Additionally, several losses are formulated with similar forms, which are cooperated for easy optimization, e.g., $L_{VAE}^x$   and $L_{VAE}^a$ ,   $L_{DIS^1}$ and  $L_{DIS^2}$ . It needs  ~45 minutes for training HSVA with 100 epochs on a single NVIDIA 1080Ti graphic card.
>
> **Q2:** More ablation studies.
>
> **A2:** Thanks for the suggestion. To show the effectiveness of these losses and model components in HSVA, besides the ablation in the Table 3, we conducted more ablation studies as follows. These results clearly show the superior results of HSVA compared with other variants. We will add these results in the revision.
>
> * Results of replacing SAD with MMD for HSVA. This indicates the adversarial approach indeed improve model generalization.
>
> | Dataset | U | S | H |
> | :-----: | :-----: | :-----: | :-----: |
> | AWA1 | 54.5 | 76.2 | 63.6 |
> | AWA2 | 51.4 | 81.9| 63.2 |
> | CUB   | 50.4 | 56.9 | 53.4 |
> | SUN   | 47.4 |	37.1	| 41.4 |
>
> * Results of replacing SAD with sliced Wasserstein discrepancy (SWD) for HSVA.
>
> | Dataset | U | S | H |
> | :-----: | :-----: | :-----: | :-----: |
> | AWA1 | 54.9 | 78.3 | 64.6 |
> | AWA2 | 53.8 | 79.6 | 64.2 |
> | CUB   | 49.8 | 58.5 | 53.8 |
> | SUN   | 46.7 | 38.5 | 42.2 |
>
> * Results of using two classifiers without $L_{DIS}$ in SA for HSVA.
>
> | Dataset | U | S | H |
> | :-----: | :-----: | :-----: | :-----: |
> | AWA1 | 54.6 | 77.2 | 64.0|
> | AWA2 | 53.1 | 80.3 | 63.9 |
> | CUB   | 50.7 |57.3	| 53.8 |
> | SUN   | 45.5 |	39.1 | 42.1 |
>
> * Results of using one classifier for representing the discrepancy in SA for HSVA.
>
> | Dataset | U | S | H |
> | :-----: | :-----: | :-----: | :-----: |
> | AWA1 | 57.0 | 74.2 | 64.5 |
> | AWA2 | 55.3 | 77.2 | 64.4 |
> | CUB   | 50.7 | 58.7 | 54.4 |
> | SUN   | 48.7 |	37.7	| 42.5 |
>
> * Results of using three classifiers for representing the discrepancy in SA for HSVA.
>
> | Dataset | U | S | H |
> | :-----: | :-----: | :-----: | :-----: |
> | AWA1 | 55.6 | 76.1 | 64.3 |
> | AWA2 | 55.1 | 81.0 | 63.0 |
> | CUB   | 50.3 | 58.7	| 54.2 |
> | SUN   | 45.6 |38.8	| 41.9 |

---

### Decision · Program_Chairs · 2021-09-27

**Decision:**

Accept (Poster)

**Comment:**

This paper presents a two-stage adaptation approach, i.e., structure adaptation (SA) and distribution adaptation (DA), for closing the gap between the visual (image) feature space and the semantic (e.g., attributes) feature space, resulting in a common space that not only captures the manifold structure but also induces the discriminative property. With the learned common space, (G)ZSL can be easily fulfilled, and very promising experimental results on four ZSL benchmarks are achieved.

This work is motivated and posed on the foundation of CADA-VAE [11]. Through the authors’ rebuttal as well as the insightful discussion among the reviewers, the AC thinks that the improvement over [11] brought by the proposed HSVA is sufficient in terms of both methodology and performance. The authors proffered a good rebuttal, where more ablation studies are shown to demonstrate the effectiveness of HSVA. Taking together all the detailed comments, the AC votes for accepting the paper and would believe that it can advance the research of ZSL.  In the camera-ready version, the authors are suggested to: 1) strengthen the writing, 2) supplement the key ablation results in the main paper, and 3) highlight the differences from the existing works and clarify the necessity of the overall objective in Eq. (13).